# COVID-19, Urbanization Pattern and Economic Recovery: An Analysis of Hubei, China

**DOI:** 10.3390/ijerph17249577

**Published:** 2020-12-21

**Authors:** Wenyi Yang, Xueli Wang, Keke Zhang, Zikan Ke

**Affiliations:** 1Institute of Regional and Urban-Rural Development, Wuhan University, Wuhan 430072, China; wenyiyang@whu.edu.cn (W.Y.); sherrywang@whu.edu.cn (X.W.); zhangkk@whu.edu.cn (K.Z.); 2School of Foreign Languages, Huazhong University of Science and Technology, Wuhan 430074, China

**Keywords:** COVID-19, urbanization pattern, economic recovery

## Abstract

In the context of the rapid development of urbanization and increasing population mobility in China, the outbreak of COVID-19 has had a significant impact on China’s economy and society. This article uses China UnionPay transaction data and takes Hubei, the worst-hit region by COVID-19 in China, as an example, to conduct empirical analysis using the generalized method of moments (GMM) of the impact of current urbanization patterns on the spread of the epidemic and economic recovery from the perspectives of time, industry, and regional differences. The study found that during the different stages of COVID-19, including discovery, outbreak, and subsidence, the overall impact of urbanization on the economy in Hubei Province was first positive, then became negative, and finally gradually increased. This process had significant industrial and urban heterogeneity, which was mainly manifested in losses in tourism and catering industries that were significantly greater than those in the audio-visual entertainment and digital office industries. Similarly, the recovery speed of large cities was lower than that of small and medium-sized cities. The main reason for these differences is that the one-sided problem of urbanization is more obvious in areas with higher urbanization rates. COVID-19 has drawn attention to the development of urbanization in the future, that is, the development path of one-sided economic resource agglomeration and scale expansion should be abandoned, with greater attention paid to the improvement of service functions and the development of amenities. This transformation is necessary to enhance urban economic resilience and reduce public health risks.

## 1. Introduction

COVID-19 broke out suddenly at the end of 2019. In the context of rapid urbanization and regional integration, the speed and scope of COVID-19 are significantly greater than previous epidemics, posing a major threat to public health and the development of the economy and society. Similar to the situation of SARS, cities with dense populations, large scales, and high mobility were again the worst-hit areas. China’s urbanization has made significant improvements during the past 40 years. The urbanization rate of the permanent population has increased from 17.92% in 1978 to 60.60% in 2019, and per capita GDP has also increased from RMB 381 to RMB 70,892. However, the rapid development of urbanization not only failed to protect China from the impact of the epidemic, but also resulted in a greater loss. For example, in Wuhan, where the epidemic was the most severe in China, during the first quarter of 2020, the total number of confirmed cases reached 50,007, GDP decreased by 40.5%, the added value of the tertiary industry decreased by 37.7%, and the total retail sales of consumer goods decreased by 45.7%. Although other cities’ situations were less serious compared to Wuhan, they also suffered significant losses. This phenomenon has led to a rethinking of urbanization. What is the relationship between urbanization, the spread of the epidemic, and economic recovery? Why has the urbanization process not improved the ability of cities to resist the crisis?

Based on the above problems, this study analyzed the specific impact of current urbanization pattern on economic recovery during the COVID-19 period from the perspectives of timing, industry classification, and regional differences. The aim was to identify the shortcomings of the current Chinese urbanization model, and provide useful suggestions for economic recovery and high-quality development of urbanization in the post-epidemic period. The structure of the article is as follows: The second section presents literature review, which separately discusses the experience of urbanization and epidemics in Western society, in addition to the background of COVID-19 in China and its impact on urbanization and the economy. The third section introduces the data and methodology. The fourth section provides the situation analysis of COVID-19, urbanization patterns, and economic development in Hubei Province of the first four months of 2020. Next, the fifth section provides analysis of the economic recovery effect of the urbanization pattern of Hubei Province under the impact of the COVID-19 based on the results of the generalized method of moments (GMM) models. Finally, the sixth section presents conclusions of the article, and puts forward several suggestions.

## 2. Literature Review and Theoretical Elaboration

### 2.1. Urbanization and Epidemics

Urbanization, in essence, is the evolution of the social and economic structure characterized by the continuous spatial agglomeration of population and production factors in the process of industrialization. This process can be divided into two stages. One is the material expansion stage, that is, the rural population transfers to non-agricultural industries contributing to the expansion of urban areas; the other is the cultural diffusion stage, which refers to the spread of urban values to the countryside [1]. From a broader perspective, urbanization is an all-round change including the economy, society, politics, culture, and population, which evolves along a curve similar to the S-shape [2]. The urbanization of Western developed countries has undergone a long process. It took 80 years for the urbanization rate to rise from 30% to 70% in the United States [3], and about one hundred years in the United Kingdom [4] and Germany [5]. Japan is the country with the fastest urbanization rate among the developed countries, and it also took nearly 50 years to achieve this milestone [6]. International experience shows that, when the urbanization rate exceeds 50%, it enters a critical growth period, and the “urban disease” gradually appears [7]. At the end of the 19th century, 70% of the British population lived in cities, making Britain the first country in the world to achieve urbanization. However, due to the rapid expansion of the urban population in Britain, housing shortages, the prevalence of worker slums, air and water pollution, and lack of public sanitation facilities caused British cities to foster various diseases and led to several cholera outbreaks. To remove the threat of disease transmission, the British government passed the “Public Health Act” in 1848, and began to carry out strong interventions on public health and other social issues, which greatly improved the national management capability and the ability to prevent public health incidents [8]. In the process of urbanization in Germany, water shortages and poor water quality also caused severe cholera outbreaks in the 1870s [9]. To address various problems in the process of urbanization, a number of theoretical approaches, such as the garden city [10], satellite city [11], neighborhood unit [12], and organic decentralization [13] theories have been proposed to improve the urbanization model.

Although China’s urbanization practice has followed the general pattern of Western urbanization, it has also shown its own characteristics. China’s urbanization started from the reform and opening up in the 1980s. After more than 40 years of development, China’s urbanization rate has increased from 17.92% in 1978 to 60.60% in 2019, which is the largest and fastest urbanization process in global history. Industrialization is the driving force for the rapid development of China’s urbanization. As a result, however, China’s urbanization has also followed an extensive development path [14], with too much attention paid to the production function and economic growth, while the development of service functions and amenities in cities has been ignored [15]. A large quantity of low-cost labor flowed from rural areas to cities, resulting in a substantial demographic dividend for China’s urbanization development. However, the contradiction between the insufficient supply of urban supporting facilities and basic public services, such as education and medical care, is becoming increasingly obvious, resulting in a series of urban problems, such as community poverty, traffic congestion, and environmental pollution [16,17,18,19]. In essence, the urban problem is the specific manifestation of the contradiction between population increase and the lagging development of urban management and service capacity [20]. Under the impact of public health emergencies, this contradiction is more prominent and aggravates the problem of epidemics in cities. The experience of the spread of the COVID-19 has deepened our understanding of the effects of urbanization, that is, urbanization has a duality in the spread of epidemics. That is, urbanized areas have high population density and mobility, making them a “hotbed” for the spread of the epidemic. At the same time, however, compared with rural areas, cities have gathered relatively richer medical resources and public facilities, which provide favorable conditions for epidemic prevention and control. Therefore, urbanization is a double-edged sword for the spread of the epidemic [21].

### 2.2. COVID-19, Urbanization and Economic Recovery

Urbanization and regional integration constitute the current background in China, leading to unprecedented complexity in the spread and prevention of the COVID-19 epidemic. Compared with SARS in 2003, the impact of COVID-19 on the economy and society is more profound. First, China’s urbanization rate in 2003 was 40.53%, and increased to 60.60% by the end of 2019. As the capital of Hubei Province, Wuhan’s urbanization rate in 2019 reached 80.49%, significantly higher than the national level. Due to the rapid development of urbanization, population density and mobility have greatly increased. In 2019, the scale of China’s population flow was as high as 236 million people, which is about six times that of 17 years ago. Baidu Migration Data shows that before Wuhan took measures to lock down the city due to COVID-19, about 1.5 million people flowed out of Wuhan to Huanggang and Xiaogan, making them the two most severely affected cities after Wuhan [22]. Secondly, from the perspective of the economy, investment and exports were the main driving forces of economic development in 2003, but now domestic consumption has become the primary driving force. In 2019, China’s final consumption expenditure contributed 57.8% to GDP, compared with 35.4% in 2003. The tertiary industry accounted for 53.9% of GDP in 2019, nearly 12 percentage points higher than in 2003. In the case of significant changes in the economic structure and driving force, the outbreak of COVID-19 directly limited the consumption behavior of the population. The subsequent decline in consumption led to stagnation of most production and service industries, causing significant economic losses in urbanized areas. The destructive impact of COVID-19 on consumption mainly occurred via a reduction of residents’ income, increasing unemployment [23], and growth in precautionary savings [24]. Nonetheless, high rates of agglomeration and mobility characteristics in cities stimulated new development opportunities during COVID-19. Due to the development of information technology and the Internet, city populations represent large markets and provide the basis for the emergence and development of new business modes for some industries. During the epidemic, online retail and fresh food e-commerce developed rapidly. Emerging business modes, such as online medical care, online education, live broadcasting, and remote offices were widely promoted. COVID-19 helped optimize people’s consumer psychology, needs, and structure. The demand for sanitary products, medical care, and sports fitness products increased significantly, and the willingness to buy insurance, entertainment, and education also improved significantly [25]. Faced with these new trends and characteristics, continuing to promote their development is an important task for cities in the post-epidemic period.

However, China’s urbanization suffers from the problem of one-sidedness in its functions, creating a spatial imbalance [26,27] and presenting hidden dangers for the recovery and development of economy in the post-epidemic period. From the perspective of urban function, although the agglomeration degree and economic efficiency of the city have steadily increased, public services and safety awareness have not improved simultaneously. Resources and funds are preferentially inclined to industries with productive benefits [28,29,30], whereas the development of consumption, medical and other public services is slow, leading to a series of problems. For example, the surge in demand for epidemic treatment has magnified the insufficiency in medical and health care, causing the medical sector to function ineffectively or even collapse. From a spatial perspective, significant regional differences exist in the level of urbanization and the supply of public services, increasing the challenges of achieving joint control of COVID-19 at a regional scale, and also adversely affecting the overall recovery of the regional economy in the post-epidemic period. The profound lessons of epidemic prevention and control indicate that China’s urbanization in the future needs to balance the expansion of scale and the improvement of internal functions [21]. It is also necessary to strengthen the cooperation among cities in joint epidemic control, resource allocation, and information sharing at the regional level, to improve cross-border management and the public services of cities.

In summary, from the experience at home and abroad, cities are not only hotbeds of epidemic diseases, but also the battlefields of epidemic prevention and control [25]. Research on urbanization and its problems in various countries provides useful references for understanding the general laws of urban development, but the issue of urbanization in China requires special attention due to its own characteristics. China’s research on urbanization and its problems mainly focus on environmental pollution [31,32,33,34,35,36,37], traffic congestion [38,39], or spatial economic differences [40,41], whereas analysis of public health emergencies in cities is insufficient. The current research on COVID-19 in China has mostly analyzed the evolution of the epidemic, or discussed its economic impact at a macro level, but has not studied its impact from the perspective of urbanization. Thus, taking Hubei Province, the region with the most COVID-19 cases in China, as an example, this study focused on three key concepts, namely, COVID-19, urbanization, and economic recovery. The study used China UnionPay transaction data to analyze the economic trends of Hubei Province under the interactive influence of the current urbanization pattern and COVID-19. The aim was to provide ideas and references for its economic recovery and high-quality urbanization in the post-epidemic period.

## 3. Data and Methodology

### 3.1. Data and Research Scope

The unique data used in this paper is China UnionPay transaction data, extracted from the China UnionPay personal bank card database, which includes detail regarding cities issuing the cards, date and time of transactions, places where cards were physically swiped, names of stores, and amount of payment. In addition to traditional debit/credit card swiping, the means of transaction also includes mobile payments, such as scan to pay via Cloud QuickPass provided by China UnionPay, but does not include WeChat and Alipay QR code payment. The data includes the total daily transaction volume and the transaction amount of 18 industries, such as catering and snacks, real estate, transportation services, and education, from January to April in 2020, covering all cities in Hubei Province (Figure 1).

The COVID-19 case data was taken from the Health Commissions of Hubei Province and cities, with details including the total number of confirmed cases, daily new cases, cured cases, and deaths from January 10 to June 5, 2020. The daily actual cases were calculated by subtracting the total cured cases and the total deaths from the total confirmed cases.

To study the economic impact of the epidemic, this study also used control variables, including population density, per capita disposable income, public financial expenditure, proportion of tertiary industry, number of Internet users, number of health institutions, and number of doctors per thousand people in Hubei Province. These variables were taken from the statistics bureaus of Hubei Province and cities. The statistical characteristics of variables are shown in Table 1.

### 3.2. Methodology

First, we analyzed the COVID-19 situation, economy, and urbanization pattern of each city in Hubei Province, and then tested the relationship between them using econometric models. In terms of model setting, COVID-19 is an exogenous shock to the economy, and the urbanization rate is an internal factor affecting the economy. Therefore, the actual cases of the city ($\alpha_{1}Actualcase_{i,t}$) and urbanization rate ($Urbanization_{i,t}$) were used as the core explanatory variables of the model to test their impact on the economy. Although all cases in Hubei Province were cured by the end of April, the epidemic continued to have an impact on the economy and people’s psychology. Therefore, the lagged term of the epidemic ($Actualcase_{i,t-1}$) was added in the model to test its long-term economic impact. Policy is another important factor that must be considered. During COVID-19, cities in Hubei Province implemented strict city lockdown policies, which greatly affected economic development and people’s behavior. Thus, we established city lockdown policy dummy variables ($Closepolicy_{i,t}$). According to the implementation of the lockdown policy, the dummy variables before and after the lockdown were set to 0, and the dummy variables for cities, with the exception of Wuhan, during the lockdown period were set to 1. Because the city lockdown policy of Wuhan was the strictest and the longest, the dummy variable of Wuhan was set to 2 to accurately analyze the impact of this policy. Finally, considering the complex relationship between the urbanization and epidemic control, an interaction term was added to the model to test the interaction effect of the two on economic development. The final model was set as follows:$$Economy_{i,t}=\alpha_{0}+\alpha_{1}Actualcase_{i,t}+\alpha_{2}Actualcase_{i,t-1}+\alpha_{3}Urbanization_{i,t}+\alpha_{4}Closepolicy_{i,t}\;+\alpha_{5}Actualcase\times Urbanization_{i,t}+\beta X_{i,t}+\gamma_{i}+\delta_{t}+\varepsilon_{i,t}$$

In the formula, *i* and *t* represent city and date respectively, $\alpha_{0}$ is the constant term, *X* is the control variable, $\gamma_{i}$ is the space fixed effect, $\delta_{t}$ is the time fixed effect, and $\varepsilon_{i,t}$ is the error term.

## 4. COVID-19, Urbanization and Economic Situation in Hubei Province

### 4.1. COVID-19 Situation Analysis

The earliest case of COVID-19 in Hubei Province can be traced back to December 8, 2019. However, due to the small number of infections and similar symptoms to flu, it did not attract attention in the initial stage. The earliest notification on COVID-19 issued by the Wuhan Municipal Health Commission was on December 31, 2019, when 27 confirmed cases were announced. Overall, from January 10 to June 5, 2020, the total number of confirmed cases of COVID-19 in Hubei Province increased from 41 to 68,135, the total number of cured cases increased from 2 to 63,623, and the total number of deaths increased from 1 to 4512. The number of actual daily cases increased from 38 to the peak of 50,633 on February 18, and then gradually reduced to 0 on April 26, thereafter entering the zeroing phase (Figure 2). In terms of the rate of change, the highest number of new cases per day was 14,840 on February 12; the highest number of new cured cases was 3203 on February 27; and the highest number of new deaths was 242 on February 12 (Figure 3).

The daily new cases were a direct reflection of the spread of the epidemic. The inflection point of new cases (February 12) means that the trend of case growth had declined. The inflection point of the actual cases (February 18) indicates that the rate of cures began to exceed the infection rate of COVID-19, that is, the stock of cases began to decline. Only the inflection point of actual cases can genuinely indicate that the overall epidemic situation has been relieved. Accordingly, the change trend of COVID-19 in Hubei Province from January 10 to June 5 can be divided into four stages. The first was a short and stable initial stage (January 10–January 24); the second was a rapid increasing stage (January 25–February 18); the third was a continuous decline stage (February 19–April 25); and the fourth was the clearing stage (April 26–June 5).

From the perspective of spatial differences among cities, Wuhan suffered the most severe impact of COVID-19 among the cities in both Hubei Province and China as a whole. As of June 5, 2020, the total number of confirmed cases in Wuhan was 50,340, which accounted for 73.88% of those of Hubei population and 60.63% of those of China; the total number of cured cases was 46,471, accounting for 73.04% of those of Hubei and 59.33% of those of China; and the total number of deaths was 3869, accounting for 85.75% of those of Hubei and 83.49% of those of China. Among other cities in Hubei Province, Xiaogan, Huanggang, Jingzhou, and Ezhou were the most severely affected cities following Wuhan. The total numbers of confirmed cases, cured cases, and deaths in Xiaogan accounted for 5.16%, 5.33%, and 2.86% of the provincial totals, respectively (Table 2).

### 4.2. Urbanization Pattern and Economic Situation

The urbanization rate of each city in Hubei Province in 2019 is shown in Figure 4. As the provincial capital, Wuhan has the highest urbanization rate of 80.49%. Ezhou is ranked second with an urbanization rate of 66.3%, and Huangshi’s urbanization rate is 61.7%. With the exceptions of Huanggang, Enshi, and Shennongjia, the urbanization rates of other cities exceed 50%. Taking Wuhan as an example, in terms of the urbanization pattern from 2004 to 2018, although its GDP, and secondary and tertiary industries showed a downward trend, they are nonetheless higher than the growth rates of doctors and hospital beds per thousand people (Figure 5). It can be seen that the focus of Wuhan’s urbanization model is economic growth and industrial development, whereas the improvement of basic public services such as medical care has relatively lagged behind. This laid a foundation for the huge losses caused by the outbreak of COVID-19.

Combining the trend of COVID-19 and the total transaction volume, it is obvious that there is a roughly negative correlation between the epidemic and economic trends (Figure 6). In the early stage of the epidemic, when there were fewer cases, the transaction volume of Hubei Province showed a trend of high growth with fluctuations. During the continuous spread of the epidemic, affected by the lockdown policy of Wuhan and other cities on January 23 and 24, the daily trading volume dropped precipitously, then remained low after a short recovery. The short-term rise in trading volume during this stage may have been caused by residents purchasing a large number of reserved materials before home isolation, and the subsequent decline was a chain reaction of reduced purchase after material reserve. Finally, when the epidemic subsided, with the unblocking of other cities in Hubei Province, with the exception of Wuhan, the daily transaction volume rebounded but fluctuated frequently. After Wuhan announced its unblocking and clearing of cases on April 8, the daily transaction volume in Hubei Province showed a significant rebound, reaching the level of transaction volume before the outbreak of COVID-19 at the end of April.

During the spread of COVID-19, there was significant industry heterogeneity in transaction trends (Figure 7). In terms of volume, from January to April in 2020, the industry with the largest transaction volume in Hubei was general department stores, followed by clothing, shoes and bags, home building materials, cultural, sports and entertainment, catering and snacks, and financial services. Medical services ranked fourth from the bottom among 18 categories, and education and training was ranked last. The highest ranked transaction amount of general department stores is consistent with the situation that residents’ consumption was mainly to meet the requirements of daily life under the condition of home isolation. Large transaction volumes in recreation, sports, and catering and snacks reflect that people spent a significant amount of their leisure time on audio-visual entertainment, reading, cooking, home fitness and other activities during the period of home isolation and suspension of work. Increased financial service transactions may be a manifestation of people’s increased demand for personal and medical insurance. The lower consumption of medical services may be a reflection of the reduced demand for medical care due to concerns about the high risk of infection in hospitals during COVID-19. The transaction volume of education and training was the lowest, reflecting the significant impact of home isolation on education and training during the epidemic period.

Finally, by comparing the total monthly transactions and cases in cities of Hubei Province, we can clearly see the differences in the spatial distribution of COVID-19 and the economy (Figure 8). In terms of the spatial distribution of transaction volumes, although COVID-19 had a significant impact on Wuhan’s economic activities, due to its larger population compared with other cities, Wuhan’s transaction volume remained the highest from January to April, reaching RMB 125.63 billion, accounting for more than one-third of the whole province. This was followed by Xiangyang, Jingzhou, Yichang, and Huanggang, whose total transaction volumes from January to April were RMB 38.402 billion, RMB 34.801 billion, RMB 27.484 billion, and RMB 24.959 billion. The total transaction volume generally presents an inclined triangle pattern, with Wuhan, Xiangyang, and Jingzhou as the three vertices. The transaction volume was high in the east, low in the west, and extremely low in other areas. The spatial distribution of total cases in Hubei Province was roughly characterized by being high in the northeast and low in the southwest, high in the north and south, and low in the middle. Thus, the spatial differences between COVID-19 and the economy reflect that differences exist between the influencing factors of epidemic spreading and economic development. In the early stage of the outbreak of the epidemic, geographical proximity was the main factor affecting the spatial distribution of cases. However, after implementing lockdown policies and home isolation, the influence of geographical proximity was reduced, and the urbanization level of cities became the key factor influencing epidemic control and economic recovery.

## 5. Analysis of the Economic Recovery Effect of Urbanization under COVID-19

### 5.1. Analysis of the Overall Impact on Urban Economy

First, the results of the ordinary least squares regression analysis (OLS) show that actual cases had a significant negative impact on the transaction volume of cities in Hubei Province, but did not inhibit the recovery of transactions in the later period. There is a significant negative correlation between the urbanization rate and the transaction volume, and the city lockdown policy also has a greater inhibitory effect on the transaction volume. The estimated results of the control variables show that per capita disposable income and the number of hospitals have a negative impact on transactions during the spread of COVID-19, whereas public financial expenditure, the proportion of tertiary industry, and Internet users have a positive effect on transactions (Table 3).

However, considering that there may be a causal endogenous relationship between the urbanization rate and the transaction volume, the results of the OLS model may be biased, thus, the instrumental variable method was used to modify the model. The birth rate was chosen as an instrumental variable of the urbanization rate because cities have implemented stricter fertility policies than in rural areas for a long period, and the cost of raising children in urban areas is higher than that in rural areas, reducing the relative willingness of urban residents to bear children. This affects the urbanization rate but has no direct relationship with the volume of urban transactions. In addition, in China, colleges and universities are often located in cities. The more developed the urban economy, the greater the number of colleges and universities; for example, there are 93 universities in Beijing and 83 universities in Wuhan. Colleges and universities attract a large number of students to the city, which has a positive impact on the urbanization rate of the permanent population. COVID-19 broke out during the school winter vacation. As a result of the lockdown policy, a large number of college students from other places were unable to return to school; consequently, they had no relationship with the transaction volume of the city during this period. Therefore, the number of college students was selected as another instrumental variable of the urbanization rate. The *p*-value of the Hausmann test was 0, indicating that the urbanization rate is the endogenous variable of the model. The two instrumental variables, population birth rate and the number of college students, have significant correlations with the urbanization rate, and pass the weak instrumental variable and exogenous test, which shows that they are effective and reasonable (Table 4).

Columns two to four in Table 3 show the results of the instrumental variable methods. The results of these three models are relatively close, indicating that the models are robust. The model results with instrumental variables are substantially different from the OLS results. First, the urbanization rate has a significant positive impact on the transaction volume, indicating that the urbanization rate is an important factor supporting economic development. The higher the urbanization rate, the greater the transaction value. Second, the number of Internet users has a negative impact on transaction volume, indicating that the Internet penetration in cities is relatively insufficient and did not effectively promote online transactions, which is not conducive to the increase in transaction volumes. Third, the influence of the number of hospitals on the transaction volume was positive, indicating that the more health institutions, the greater the benefit to economic development. This result is closer to actual conditions and expectations.

Overall, the model results preliminarily show that COVID-19 was the main factor hindering economic development, and the level of urbanization was a favorable condition to support economic development. Specifically, in the period of the epidemic, the negative impact of urbanization was manifested in the fact that there were more confirmed cases in urban areas, and strict isolation and control measures were implemented in cities, which resulted in fewer trading activities; the reduction of per capita disposable income also reduced the consumption of urban residents; and the lack of Internet infrastructure hindered the development of online trade. In contrast, the positive impact of urbanization was manifested in the fact that cities had greater public financial expenditure and medical facilities, which are favorable conditions for economic regulation and epidemic control. In addition, a higher level of tertiary industry in cities is important for maintaining economic transactions.

### 5.2. Analysis from the Perspective of Different Time Periods

To further explore the impact of urbanization on the economy at different stages of COVID-19, we divided the period from January to April 2020 into three stages for GMM model estimation (Table 5). The three phases are the initial phase (January 10–January 23), the peak phase (January 24–February 18), and the decline phase (February 19–April 30). Because cities did not implement a lockdown policy in the initial stage, and all cities implemented this policy in the peak stage, cities only experienced a policy change from lockdown to unblocking during the third stage, thus, the dummy variable of lockdown policy was only added to the third stage model. In addition, the interaction term of urbanization and COVID-19 was introduced into the model and compared with the results in Table 3.

From the perspective of the whole stage, the actual cases and the lockdown policy had significant negative impacts on urban economy. The GMM results in Table 3 show that there is a positive correlation between the urbanization rate and the economy, whereas the interaction term of the urbanization rate and actual cases in Table 5 had a negative impact on the economy and did not pass the significance test. This shows that the positive effect of urbanization on economy will decrease and be no longer significant when the impact of COVID-19 is considered. Regarding the first stage, due to the small number of cases in this stage, the actual cases did not show an inhibitory effect on the economy, but because of the gradual increase in cases, there is a positive correlation with the increase in transaction volume. Under the influence of the epidemic, the urbanization rate had a negative impact on the economy. From the specific aspects of urbanization, population density and income levels had negative impacts on economy at the beginning of the epidemic, whereas the impacts of public finance and Internet users on the economy were not significant. The contradiction between supply and demand of medical services in the early stage had not yet increased, which showed a positive influence on the economy.

In the second stage, the epidemic gradually developed to its peak, but the increase in cases was not the main factor resulting in the decline of economy. Instead, the two showed a significant positive relationship. The rapid spread of the epidemic caused the urbanization rate to have an adverse impact on economic development. From the specific aspects of urbanization, the higher the population density, the greater the decline in transaction volume. The per capita disposable income and the number of Internet users had no significant impacts on the economy, whereas the outbreak of the epidemic highlighted the conflicts of demand and supply in medical care, which had a negative impact on the economy. The tertiary industry level had a positive impact on economy.

In the third stage of the epidemic’s subsidence, the impact of actual cases on the economy was no longer significant, and the lockdown policy was the main factor restraining the economy. At this stage, the number of cases decreased to zero, causing the negative impact of the urbanization rate on the economy to decline and be no longer significant. From the specific aspects of urbanization, population density had no impact on the economy, whereas the reduction in income levels had become a major obstacle to economic recovery. At this stage, the contradiction between medical supply and demand declined, which changed its impact on the economy from negative to positive, and the impact of Internet users on the economy also turned positive.

It can be seen that the epidemic did not reduce market demand, and the adverse impact on the economy came mainly from the lockdown and isolation policies, the decline in per capita income, and the lack of urban medical facilities and Internet facilities. In contrast, the city’s public financial expenditure and the level of the tertiary industry were positive factors for stabilizing the economy during the epidemic, and also had a significant impact on economic recovery in the post-epidemic period.

### 5.3. Analysis from the Perspective of Different Industries

From the previous analysis of the economic situation, we can see that significant differences existed in the industrial development situation during COVID-19. In this regard, we analyzed the heterogeneous impact of the epidemic and urbanization on the economy from the perspective of the industry (Table 6). 

Overall, the negative impacts of the epidemic and lockdown policy on industries only existed in the short term and will not hinder long-term development. Under the influence of COVID-19, the urbanization rate had a relatively smaller negative impact on most industries, while still maintaining a positive impact on six industries. The negative effects of urbanization were mainly manifested in industries such as catering and snacks, clothing, shoes and bags, home building materials, residential services, maternal and child parenting, and general department stores. The lockdown policy, the decline in per capita income, and insufficient medical supplies were the main sources of negative effects, and their negative effects exceeded the supporting effects of public financial expenditure and the level of the tertiary industry on economy. 

During the epidemic, the positive effects of urbanization mainly existed in the real estate, public services, education and training, financial services, and digital office industries. This reflects the fact that people’s demand for these services did not decrease during the epidemic, and the areas with higher urbanization levels can better meet these needs of residents due to their rich public service facilities and advanced education and financial level, thus maintaining the development of these industries. However, the urbanization rate has a negative impact on the development of cultural, sports, entertainment, and leisure tourism. This reflects the fact that cities in Hubei Province have significant deficiencies in leisure and entertainment, which limited residents’ consumption and was not conducive to the recovery of these industries in the post-epidemic period.

### 5.4. Comparative Analysis of Wuhan and Other Cities in Hubei Province

The fact that Wuhan had the greatest concentration of COVID-19 cases and implemented the lockdown policy for the longest time might influence our judgment of the true situation in other cities. Therefore, we separately estimated Wuhan and other cities to explore the spatial differences in the economic recovery effect of their urbanization patterns.

It can be seen from Table 7 that the urbanization rate of Wuhan had a significant negative impact on other industries, with the exception of general department stores, during the period when the epidemic subsided. On the contrary, the urbanization rates of other cities had a positive impact on most industries, and only had adverse impacts on the development of real estate, public services, financial services, and others. Regarding the city lockdown policy, its impacts on Wuhan’s twelve industries were not significant, whereas it had a significant negative impact on most industries in other cities. From an industrial perspective, general department stores suffered the least because of the relatively small elasticity of consumer demand for daily necessities. However, the real estate, financial services, and medical services industries suffered significant losses. Public services and medical care were also significantly impacted, which was mainly due to the extreme shortage of public services and medical resources in Wuhan during COVID-19, in addition to the reduced demand of residents in order to avoid infection.

The significant difference between Wuhan and other cities in the province indicates that there was no stable positive relationship between the urbanization rate and the ability of economic recovery. The economic recovery of Wuhan in the post-epidemic period was not as strong as that of other cities. The severity of COVID-19 in Wuhan was significantly greater than that of other cities, placing substantial pressure on epidemic control and delaying the resumption of work and production, thus resulting in the greatest economic loss in Wuhan. In addition, Wuhan’s population density, mobility, and the proportion of tertiary industry were also much higher than those of other cities, which in turn exacerbates the economic losses Wuhan suffered during the epidemic. For other cities, the prevention and control pressure due to COVID-19 was relatively light, and the economic losses were mainly caused by the city lockdown policy. Once the epidemic entered the period of subsidence, the resumption of work and production was able to be carried out as soon as possible, and the psychological expectations of residents could rebound quickly. These factors enabled other cities to recover faster than Wuhan.

The empirical analysis indicates that a higher urbanization rate is more likely to have a greater negative effect under the impact of public health emergencies. While increasing density and mobility to generate economic benefits, cities should pay more attention to the potential high risks associated with the process. The levels of urban public services and infrastructure are important factors to resist risks and promote recovery. However, in the past, urban development in Hubei Province often focused on the agglomeration of economic resources and ignored the improvement of service functions, causing the high density of urbanized areas to be accompanied by high vulnerability, and exacerbating the risks and losses caused by public health emergencies.

## 6. Conclusions and Suggestion

The rapid development of China’s urbanization has led to significant achievements, but the ability to resist public health emergencies has not improved. The outbreak of COVID-19 has a serious impact on China’s economy and society. This paper takes Hubei Province, the worst-hit region by COVID-19 in China, as an example to conduct a series of empirical analyses on the relationship between urbanization and the economy under the exogenous impact of a public health emergency.

The results highlight two dimensions of the economic effects of urbanization. The large size and high mobility of population exacerbated the spread of the epidemic and resulted in greater economic losses; conversely, the relatively developed tertiary industry and abundant public financial investment provided favorable conditions for stabilizing the economy. As the COVID-19 evolved at different stages, the overall impact of urbanization on the economy was first positive, then became negative, and finally gradually increased. During the epidemic, there was significant industrial heterogeneity in the urbanized areas. General department stores, cultural and sports entertainment, digital offices, etc., suffered relatively smaller losses, but the insufficiency and backwardness of Internet facilities and service levels restricted the development of online transactions in these industries. The current urbanization pattern suffers from the problem of non-synchronization of scale and service level, which increased the impact of COVID-19 on the economy and associated losses. As a result, Wuhan, the city with the highest urbanization rate in Hubei Province, has not recovered as fast as other cities. The reason for this phenomenon is that the one-sided problem of urbanization is more obvious in areas with higher urbanization rates. Under the sudden impact of public health emergencies, cities with high rates of urbanization face high risks and losses if they have only a large scale and high mobility, without sufficient public facilities and service levels. COVID-19 provides insights into the development of urbanization in the future, that is, the path of one-sided pursuit of economic resource agglomeration and scale expansion should be abandoned, and greater attention paid to the improvement of service functions and amenities. The latter of these is necessary to enhance urban economic resilience and improve risk prevention.

Based on the above conclusions and analysis, we propose the following ideas and suggestions for the economic recovery and high-quality development of urbanization in the post-epidemic period in Hubei Province. First, from the perspective of timing, Hubei Province and its cities should consider both short-term and long-term benefits when formulating and implementing policies. At the early stage of recovery, short-term stimulus policies, such as consumption vouchers, can be used to slow down economic losses. In the long run, it is necessary to optimize the allocation of financial resources and develop amenities to improve urban service functions and promote the upgrade of urban quality.

Second, Hubei Province must pay attention to regional differences and introduce reasonable policies to promote economic recovery in accordance with the actual conditions of each city. For Wuhan, the focus should be on promoting the momentum of new business models emerging during the epidemic, actively using technological advantages to improve infrastructure construction, and increasing the supply of amenities in public services to play a leading role in the economic upgrade of the province. For other cities, an appropriate approach is to develop specialized trading markets based on the comparative advantages of local industries and build consumer brands with regional characteristics to enhance their economic strength. In addition, these cities should also pay attention to the investment and construction of public service facilities to avoid a mismatch between production functions and service functions.

Third, differentiated economic promotion plans should be formulated based on industry heterogeneity. It is necessary to seize opportunities to promote the development of digital industries, such as e-commerce and 5G technology-based information service industries, and promote the development of emerging industries, such as remote offices, webcasting, and online education. For traditional industries, measures including reducing taxes and lowering loan interest rates can be used in the short term to reduce the financial pressure faced by enterprises. In the long term, it is important to support enterprises to promote the development of online business.

Fourth, it is necessary to promote the transformation of the concept of urban development from growth-oriented to people-oriented. In the current context of the continuous development of regional integration, population flows are more frequent, which not only increases the difficulty of urban governance, but also places higher demand on urban service levels. The insight arising from COVID-19 into the development of urbanization includes not only the importance of improving the urban service function, but also improving the collaborative governance capabilities of cities at the regional scale. Cities in Hubei Province must first improve their own service functions, and then reduce the spatial differences in the level of urbanization to lay the foundation for coordinated urban governance. Ultimately, through the improvement of collaborative governance capabilities among cities, people’s lives and property safety can be better maintained.

## Figures and Tables

**Figure 1 ijerph-17-09577-f001:**
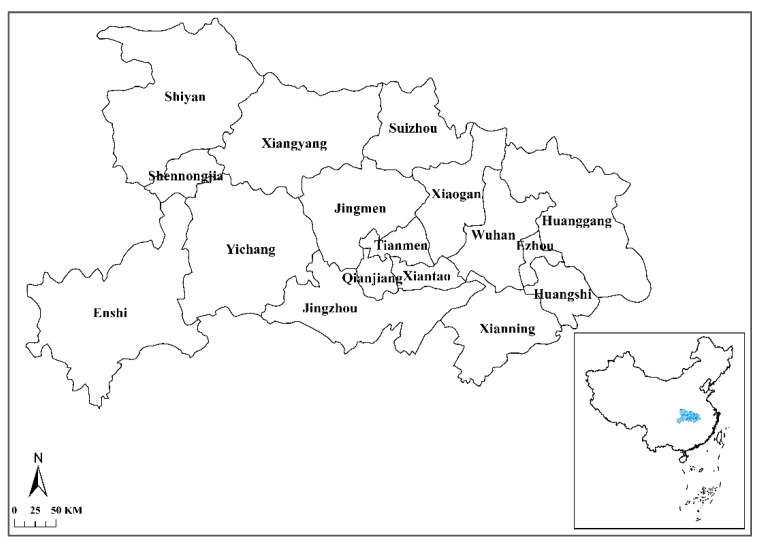
Location and administrative divisions of Hubei Province.

**Figure 2 ijerph-17-09577-f002:**
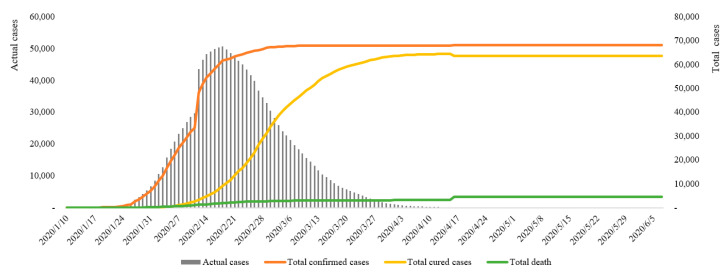
Total and actual cases in Hubei Province from January 10 to June 5.

**Figure 3 ijerph-17-09577-f003:**
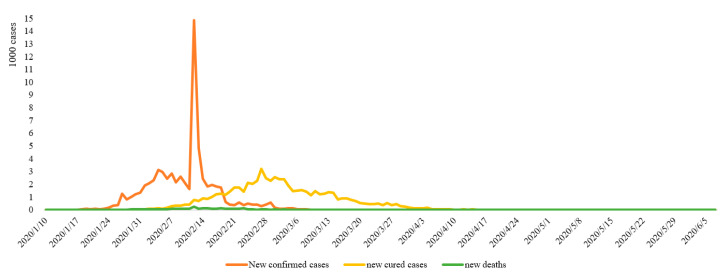
Daily trends of COVID-19 in Hubei Province.

**Figure 4 ijerph-17-09577-f004:**
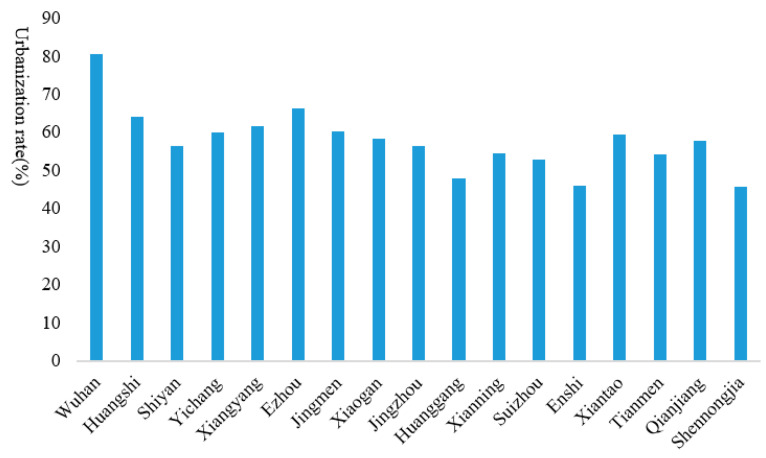
Urbanization rates of cities in Hubei Province.

**Figure 5 ijerph-17-09577-f005:**
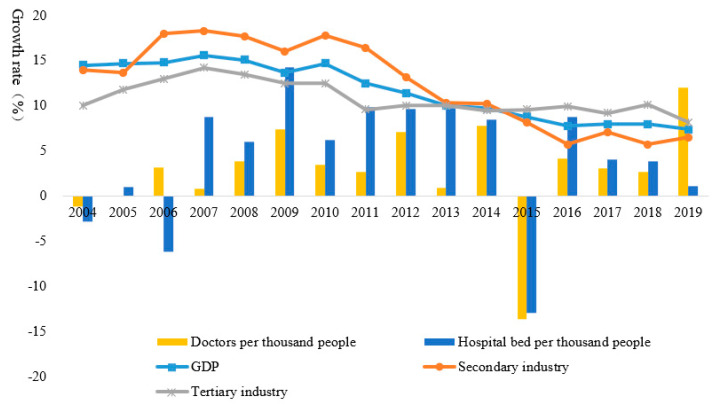
Urbanization pattern in Wuhan.

**Figure 6 ijerph-17-09577-f006:**
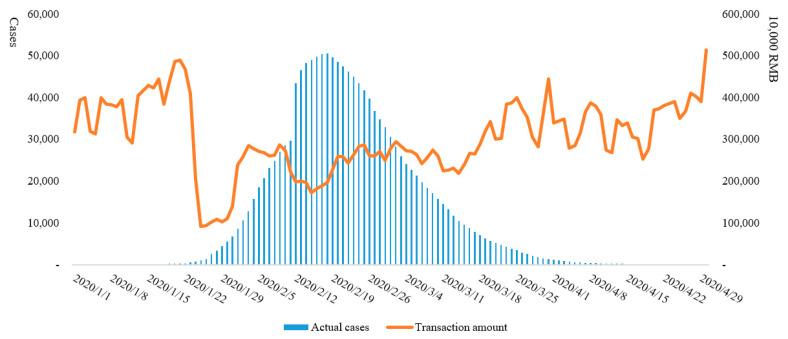
COVID-19 and economic trends in Hubei Province.

**Figure 7 ijerph-17-09577-f007:**
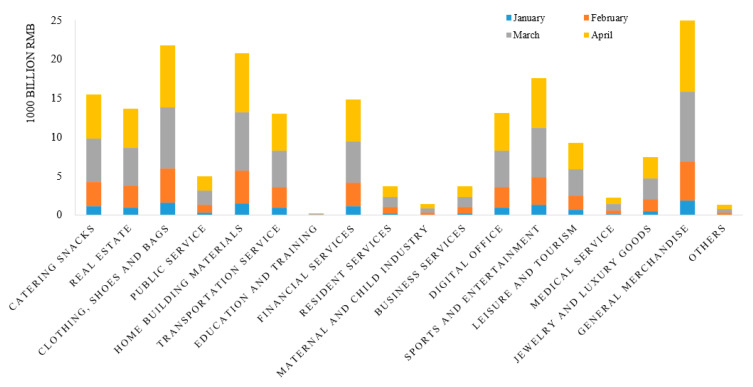
Trade volume by industry in Hubei Province.

**Figure 8 ijerph-17-09577-f008:**
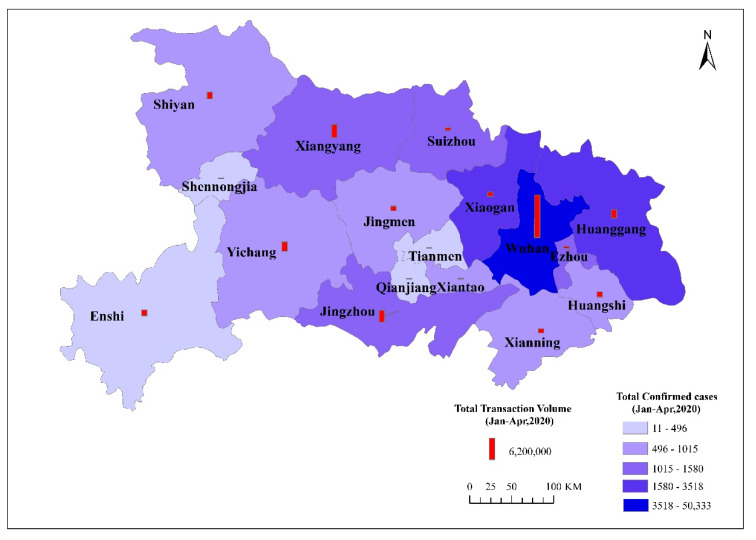
Spatial distribution of total cases and transactions in cities of Hubei Province.

**Table 1 ijerph-17-09577-t001:** Statistical characteristics of variables.

Variables		Mean	Standard Deviation	Minimum	Maximum
Case variables	Total cases	2892.57	9632.16	0	50,333
	Actual cases	741.46	3768.09	0	38,020
	Cured cases	2015.73	7450.80	0	47,283
	Deaths	135.37	530.4613	0	3869
Transaction variables	Total transaction	18,151.36	25,253.26	0.0623	21,3770
	Catering snacks	1334.70	1506.90	0	10,903
	Real estate	653.47	1968.90	0	39,491
	Clothing, shoes and bags	1251.11	4151.62	0	69,787
	Public Service	345.28	925.13	0	12,226
	Home building materials	1943.85	2115.35	0	14,629
	Transportation Service	1662.52	2633.52	0	17,468
	Education and training	47.26	270.04	0	4627
	Financial Services	1333.37	2104.44	0	22,813
	Resident services	396.29	511.73	0	3892
	Maternal and child industry	111.26	114.68	0	738
	Business services	439.96	542.93	0	4366
	Digital office	935.37	1093.66	0	8359
	Sports and entertainment	1543.56	1924.17	0	14,017
	Leisure and tourism	842.86	1006.41	0	7054
	Medical service	165.31	297.64	0	3444
	Jewelry and luxury goods	683.49	885.93	0	6543
	General merchandise	3706.72	7552.78	0	40,669
	Others	754.99	1072.48	0	8208
control variable	City lockdown policy	0.60	0.57	0	2
	Urbanization rate	57.99	8.10	45.69	80
	Population density	440.74	273.95	23.85	1308.41
	Disposable income per capita	26,492.15	6185.78	18,037	46,010
	Public expenditure	40.79	51.04	2.11	223.71
	The proportion of tertiary industry	45.44	6.37	35.2	60.8
	Number of Internet users	109.76	120.70	2.37	532
	Number of hospitals	21.71	15.95	0.86	64.97
	Number of doctors per thousand	5.57	1.43	3.64	7.56

Notes: The COVID-19 case data were taken from the Health Commissions of Hubei Province and cities; the transaction data were taken from China UnionPay; and the control variables were taken from the statistical yearbooks of Hubei Province and each city.

**Table 2 ijerph-17-09577-t002:** Rankings of total confirmed cases in the cities of Hubei Province.

Ranking	Cities	Total Confirmed Cases		Cities	Total Cured Cases		Cities	Total Deaths	
		Cases	Proportion		Cases	Proportion		Cases	Proportion
1	Wuhan	50340	73.88	Wuhan	46471	73.04	Wuhan	3869	85.75
2	Xiaogan	3518	5.16	Xiaogan	3389	5.33	Xiaogan	129	2.86
3	Huanggang	2907	4.27	Huanggang	2782	4.37	Huanggang	125	2.77
4	Jingzhou	1580	2.32	Jingzhou	1528	2.40	Ezhou	59	1.31
5	Ezhou	1394	2.05	Ezhou	1335	2.10	Jingzhou	52	1.15
6	Suizhou	1307	1.92	Suizhou	1262	1.98	Suizhou	45	1.00
7	Xiangyang	1175	1.72	Xiangyang	1135	1.78	Jingmen	41	0.91
8	Huangshi	1015	1.49	Huangshi	976	1.53	Xiangyang	40	0.89
9	Yichang	931	1.37	Yichang	894	1.41	Huangshi	39	0.86
10	Jingmen	928	1.36	Jingmen	887	1.39	Yichang	37	0.82
11	Xianning	836	1.23	Xianning	821	1.29	Xiantao	22	0.49
12	Shiyan	672	0.99	Shiyan	664	1.04	Xianning	15	0.33
13	Xiantao	575	0.84	Xiantao	553	0.87	Tianmen	15	0.33
14	Tianmen	496	0.73	Tianmen	481	0.76	Qianjiang	9	0.20
15	Enshi	252	0.37	Enshi	245	0.39	Shiyan	8	0.18
16	Qianjiang	198	0.29	Qianjiang	189	0.30	Enshi	7	0.16
17	Shennongjia	11	0.02	Shennongjia	11	0.02	Shennongjia	0	0.00

Notes: The COVID-19 case data was were taken from Health Commissions of Hubei Province and cities, and the proportion was calculated by dividing the value of cases in each city by the total value of cases in Hubei Province.

**Table 3 ijerph-17-09577-t003:** Results of the overall impact on the urban economy.

Variables	Model 1 OLS	Model 2 2SLS	Model 3 LIML	Model 4 GMM
Actual cases	−0.5283 *** (0.0863)	−0.5207 *** (0.0926)	−0.5207 *** (0.0926)	−0.5204 *** (0.0926)
Actual casest-1	0.5082 *** (0.0872)	0.5251 *** (0.0937)	0.5251 *** (0.0937)	0.5251 *** (0.0937)
Urbanization rate	−0.9399 ** (0.3972)	11.7025 *** (1.2001)	11.7070 *** (1.2005)	11.7243* ** (1.1990)
Lockdown Policy	−0.4136 *** (0.0531)	−0.4972 *** (0.0588)	−0.4972 *** (0.0588)	−0.4999 *** (0.0583)
Population density	−0.0120 (0.0263)	−0.0691 ** (0.0303)	−0.0691 ** (0.0303)	−0.0749 *** (0.0266)
Disposable income per capita	−0.9482 *** (0.2911)	−9.1606 *** (0.8245)	−9.1635 *** (0.8247)	−9.1813 *** (0.8228)
Public expenditure	0.1467 *** (0.0382)	0.0158 (0.0446)	0.0157 (0.0446)	0.0164 (0.0445)
The proportion of tertiary industry	1.4057 *** (0.0512)	1.6295 *** (0.0639)	1.6296 *** (0.0639)	1.6341 *** (0.0627)
Number of Internet users	0.0811 *** (0.0287)	−0.1649 *** (0.0402)	−0.1650 *** (0.0402)	−0.1673 *** (0.0397)
Number of hospitals	−0.0131 (0.0347)	0.2741 *** (0.0467)	0.2742 *** (0.0467)	0.2714 *** (0.0461)
Constant	19.1895 *** (1.5134)	51.9408 *** (3.6037)	51.9525 *** (3.6047)	52.1043 *** (3.5790)
R-squared	0.8863	0.8418	0.8418	0.8416

Note: Standard errors are shown in parentheses; *** *p* < 0.01, ** *p* < 0.05.

**Table 4 ijerph-17-09577-t004:** Endogeneity test and validity of instrumental variables.

Test Aims	Test Methods	Indexes	Statistics	*p*-Value
Whether there are endogenous variables	Hausmann test	chi2(1)	207.42	0.00
		Robust DWH test for heteroscedasticity	235.54	0.00
Correlation test of instrumental variables	Results of the first stage	Population birth rate	−0.05	0.00
		College students	0.03	0.00
		F-statistics	294.89	0.00
Weak instrumental variable test	Weak instrumental variable test	Minimum eigenvalue statistics	221.79	
		Cragg–Donald Wald F statistics	264.75	
		Kleibergen–Paap Wald F statistics	497.78	
	Critical value of weak instrumental variable	Critical value of 10%	19.93	
		Critical value of 15%	11.59	
Exogenous test of instrumental variables	Overidentification test	Hansen J statistics	1.963	0.161

**Table 5 ijerph-17-09577-t005:** Results of the impact on the urban economy at different stages.

Variables	Whole Stage	First Stage	Second Stage	Third Stage
Actual cases	−0.4489 *** (0.0873)	0.6681 *** (0.1082)	0.8525 *** (0.1541)	−0.1009 (0.0957)
Actual casest-1	0.4898 *** (0.0847)	−0.0693 (0.0569)	−0.0126 (−0.1263)	0.1029 (0.0845)
Lockdown policy	−0.3819 *** (0.0573)			−0.2168 *** (0.0411)
Urbanization×Cases	−0.0011 (0.0007)	−0.0097 *** (0.0016)	−0.0085 *** (0.0020)	−0.0007 (0.0005)
Population density	−0.0234 (0.0264)	−0.0376 (0.0500)	−0.2615 *** (0.0656)	−0.0141 (0.0198)
Disposable income per capita	−1.3219 *** (0.1710)	−1.0751 *** (0.2503)	−0.0173 (0.6262)	−1.0803 *** (0.1391)
Public expenditure	0.1645 *** (0.0348)	0.1406 (0.0943)	0.6639 *** (0.0794)	0.1509 *** (0.0295)
The proportion of tertiary industry	1.3692 *** (0.0501)	1.3970 *** (0.0860)	0.9858 *** (0.0973)	1.1879 *** (0.0460)
Number of Internet users	0.0908 *** (0.0303)	0.2298 *** (0.0420)	−0.0813 (0.0676)	0.0597 ** (0.0260)
Number of hospitals	−0.0020 (0.0303)	−0.0669 (0.0667)	−0.1975 *** (0.0749)	0.0646 ** (0.0261)
Constant	19.1953 *** (1.8536)	16.3458 *** (2.6790)	5.0365 (6.6163)	17.1300 *** (1.5089)
R-squared	0.8868	0.9496	0.8746	0.9354

Note: Standard errors are shown in parentheses; *** *p* < 0.01, ** *p* < 0.05.

**Table 6 ijerph-17-09577-t006:** Results of the impact on different industries.

Industries	Actual Cases	Actual Casest-1	Lockdown Policy	Urbanization × Cases	R-Squared
Catering snacks	−0.1780 ** (0.0835)	0.4950 *** (0.0763)	−0.2427 *** (0.0540)	−0.0049 *** (0.0007)	0.8730
Real estate	−2.1312 *** (0.1499)	0.7441 *** (0.1346)	−1.5404 *** (0.1040)	0.0180 *** (0.0011)	0.8237
Clothing, shoes and bags	−0.3240 *** (0.0882)	0.5142 *** (0.0811)	−0.4244 *** (0.0644)	−0.0028 *** (0.0006)	0.8754
Public Service	−2.4411 *** (0.1584)	0.7684 ** (0.1284)	−1.2574 *** (0.1113)	0.0236 *** (0.0015)	0.7196
Home building materials	−0.4557 *** (0.1033)	0.5700 *** (0.0936)	−0.2987 *** (0.0586)	−0.0020 *** (0.0007)	0.8466
Transportation Service	−0.7172 *** (0.1038)	0.6782 *** (0.0922)	−0.6424 *** (0.0665)	0.0014 * (0.0008)	0.8565
Education and training	−1.0708 *** (0.1124)	0.2541 *** (0.0885)	−0.7803 *** (0.0688)	0.0136 *** (0.0011)	0.7055
Financial Services	−0.3990 *** (0.0893)	−0.0855 (0.0806)	−0.0750 (0.0587)	0.0034 *** (0.0007)	0.8676
Resident services	−0.2162 ** (0.0858)	0.3734 *** (0.0741)	−0.3402 *** (0.0620)	−0.0024 *** (0.0007)	0.8487
Maternal and child industry	−0.1755 ** (0.0825)	0.4579 *** (0.0717)	−0.2392 *** (0.0562)	−0.0041 *** (0.0007)	0.8146
Business services	−0.4324 *** (0.0832)	0.4999 *** (0.0707)	−0.3811 *** (0.0592)	−0.0009 (0.0007)	0.8275
Digital office	−0.6721 *** (0.0974)	0.4818 *** (0.0888)	−0.2683 *** (0.0711)	0.0014 * (0.0007)	0.8205
Sports and entertainment	−0.0180 (0.0831)	0.3934 *** (0.0732)	−0.2013 *** (0.0586)	−0.0058 *** (0.0007)	0.8723
Leisure and tourism	−0.1272 (0.0869)	0.4668 *** (0.0769)	−0.2459 *** (0.0580)	−0.0055 *** (0.0007)	0.8627
Medical service	−0.7132 *** (0.0700)	0.4147 *** (0.0533)	−0.5505 *** (0.0534)	0.0046 *** (0.0008)	0.8591
Jewelry and luxury goods	−0.1010 (0.0939)	0.4606 *** (0.0823)	−0.2988 *** (0.0661)	−0.0055 *** (0.0008)	0.8525
General merchandise	−0.3488 *** (0.0947)	0.5891 *** (0.0850)	−0.2514 *** (0.0697)	−0.0039 *** (0.0009)	0.8661
Others	−0.8374 *** (0.1579)	0.8057 *** (0.1508)	−0.4422 *** (0.0915)	−0.0036 *** (0.0011)	0.7293

Note: Standard errors are shown in parentheses; *** *p* < 0.01, ** *p* < 0.05, * *p* < 0.1.

**Table 7 ijerph-17-09577-t007:** Results of the impact on different industries in Wuhan and Other cities.

Industries	Wuhan		Other cities	
	Urbanization × Cases	Lockdown Policy	Urbanization × Cases	Lockdown Policy
Catering snacks	−0.0416 *** (0.0059)	−0.0.0213 (0.0240)	0.0045 *** (0.0005)	−0.9269 *** (0.1685)
Real estate	−0.2532 *** (0.0371)	−0.7627 *** (0.0979)	−0.0031 *** (0.0008)	−1.7599 *** (0.2509)
Clothing, shoes and bags	−0.0761 *** (0.0082)	−0.1529 *** (0.0277)	0.0045 *** (0.0006)	−1.0106 *** (0.1759)
Public Service	−0.2712 *** (0.0388)	−0.0211 (0.1549)	−0.0025 *** (0.0007)	−1.3968 *** (0.2096)
Home building materials	−0.0472 *** (0.0078)	−0.0496 (0.0334)	0.0040 *** (0.0006)	−0.9214 *** (0.1788)
Transportation Service	−0.0622 *** (0.0106)	−0.1483 *** (0.0362)	0.0039 *** (0.0007)	−1.2565 *** (0.2187)
Education and training	−0.0976 *** (0.0137)	0.0456 (0.0378)	0.0020 *** (0.0005)	−1.2839 *** (0.1513)
Financial Services	−0.1895 *** (0.0398)	−0.2162 ** (0.0950)	−0.0020 *** (0.0006)	−0.4188 ** (0.1985)
Resident services	−0.0386 *** (0.0075)	−0.0095 (0.0250)	0.0049 *** (0.0006)	−1.1668 *** (0.1889)
Maternal and child industry	−0.0239** (0.0100)	0.0199 (0.0332)	0.0046 *** (0.0005)	−0.9964 *** (0.1631)
Business services	−0.0624 *** (0.0093)	−0.0399 (0.0339)	0.0042 *** (0.0005)	−1.1365 *** (0.1694)
Digital office	−0.0561 *** (0.0144)	−0.0789 * (0.0478)	0.0018 *** (0.0006)	−0.6389 *** (0.1866)
Sports and entertainment	−0.0283 *** (0.0084)	−0.0146 (0.0301)	0.0059 *** (0.0006)	−1.1574 *** (0.1934)
Leisure and tourism	−0.0432 *** (0.0073)	−0.0266 (0.0302)	0.0050 *** (0.0006)	−1.0937 *** (0.1821)
Medical service	−0.1042 *** (0.0184)	−0.1861 *** (0.0595)	0.0020 *** (0.0006)	−0.8182 *** (0.1813)
Jewelry and luxury goods	−0.0306 *** (0.0073)	0.0205 (0.0276)	0.0063 *** (0.0007)	−1.2874 *** (0.2216)
General merchandise	0.0027 (0.0089)	0.0389 (0.0321)	0.0040 *** (0.0007)	−0.9761 *** (0.2183)
Others	−0.2731 *** (0.0517)	−0.0211 (0.1260)	−0.0030 *** (0.0007)	−0.2644 (0.2098)

Note: Standard errors are shown in parentheses; *** *p* < 0.01, ** *p* < 0.05, * *p* < 0.1.

## Data Availability

Restrictions apply to the availability of these data. Data was obtained from China UnionPay and are available from authors with the permission of China UnionPay.

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
