# Peer review of "COVID-19, Urbanization Pattern and Economic Recovery: An Analysis of Hubei, China"

_ijerph, 2020, doi:10.3390/ijerph17249577_

Round 1
Reviewer 1 Report
Thank you for the opportunity to review such an important and current article.
Overall it is interesting and discusses some important issues related to urbanisation, economics and pandemics. Below are my comments:
Abstract
It needs to explain the methods used. Only stating ‘empirical analysis’ is insufficient to guide the readers in regard to what the article does. Also perhaps add the type economic analysis (the method) as a keyword.
Literature review
The article presents at times results too early. For instance, lines 138-141 show results which shouldn’t be included in the literature review. I suggest the authors carefully read the text and make sure the content of each section is appropriately presented.
Methodology
How did the researcher(s) have access to the payment data? Were there ethics concerns involved or is the analysed data freely available? Are there ethical issues involved in accessing and reporting on this data? The dataset used should be linked to a source, perhaps a website could be linked or added in the references so readers can access the sorce.
Also, in regard to the comment above, what are the sources for Tables 1 and 2?
General comments
Considering this is a health-based journal, and not a method-specific one, the paper would benefit from making the methodology and analysis clearer to a wider audience. It needs further explanation of terms, meanings and methods of analysis.
Overall the English language needs revision and there are typos in the manuscript.
This article is an important contribution to the area of urban planning and public health, however, I suggest the authors address the notes above before proceeding with the publication.
Author Response
Overall, the manuscript has been revised to address the concerns raised by the anonymous reviewers. The specific comments and changes are listed as follows in a point-to-point manner:
(1) Abstract: It needs to explain the methods used. Only stating ‘empirical analysis’ is insufficient to guide the readers in regard to what the article does. Also perhaps add the type economic analysis (the method) as a keyword.
Reply:
Thanks for the suggestion. We have changed the “empirical analysis” to “generalized method of moments (GMM)”, the special method we used in the abstract.
(2) Literature review: The article presents at times results too early. For instance, lines 138-141 show results which shouldn’t be included in the literature review. I suggest the authors carefully read the text and make sure the content of each section is appropriately presented.
Reply:
Thanks for the suggestion. We have deleted the sentences on lines 138-140.
(3) Methodology: How did the researcher(s) have access to the payment data? Were there ethics concerns involved or is the analysed data freely available? Are there ethical issues involved in accessing and reporting on this data? The dataset used should be linked to a source, perhaps a website could be linked or added in the references so readers can access the sorce. Also, in regard to the comment above, what are the sources for Tables 1 and 2?
Reply:
We are very sorry that the transaction data of China UnionPay are non-public. We obtained these data through signing a cooperation agreement with China UnionPay, for the purpose of scientific research and paper publication only. According to the agreement, we cannot provide these data to third parties without authorization. Besides, we follow your suggestions and add notes below Table 1 and Table 2 to introduce the data source and calculation method. Just as follows:
Bellow the Table 1, we added:
Notes: The COVID-19 case data comes from Health Commissions of Hubei Province and cities; the transaction data comes from China UnionPay; and control variables come from the statistical yearbooks of Hubei Province and each city.
Bellow the Table 2, we added:
Notes: The COVID-19 case data comes from Health Commissions of Hubei Province and cities, and the proportion is calculated by dividing the value of cases in each city by the total value of cases in Hubei Province.
(4) General comments: Considering this is a health-based journal, and not a method-specific one, the paper would benefit from making the methodology and analysis clearer to a wider audience. It needs further explanation of terms, meanings and methods of analysis. Overall, the English language needs revision and there are typos in the manuscript.
Reply:
Thanks a lot for the valuable suggestion. We have supplemented and modified the empirical analysis part to strengthen the explanation of methods and terminology, and carefully checked the grammar in the full manuscript to correct errors and inappropriate expressions. We believe that the language of the article has been greatly improved after modified and is easy to understand.

Reviewer 2 Report
This is a very interesting and good work. It focuses on a very important geographical case study area, as concerns the Covid-19 pandemic: the Hubei province, the area where the pandemic started. In particular, it focuses on the role of urban areas in the economic recovery of all the regions part of Hubei province, conducting several quantitative analyses based on breakdown by a space, time and industry.
I have appreciated it a lot.
The literature review, I mean the empirical and theoretical framework, for example, is very well done. The case of the Chinese urbanization is well placed in the historical context of the urban dynamics at the global scale, and the main urban issues which arose with the outbreak of the pandemic are very well presented and explained. Author took into consideration a wide and relevant literature.
The methodological approach is clear and rigorous enough, and the paper is rich of robust quantitative analyses, which are very well and clearly introduced, presented, explained, and commented. I have also appreciated the descriptive analyses: very clear and coherent.
Finally, I would say that I have been fully convinced by the analyses conducted, and by the conclusions drawn at the end. This paper is very good and very convincing.
I think that this an interesting approach, that can be fruitfully used and replied in other countries which experienced the Coid-19 pandemic: for example, in Europe.
Lastly, I would like to point out only three things:
1) Line 157: are three or two keywords? It seems that they are two: urbanization and economic recovery.
2) How many cities, and how many regions are part of the Hubei province? Therefore, I wonder: how many cities are used as spatial units in the database? Commenting on table 2 (lines 232-239), authors talk about cities and regions as if they are the same spatial units. Are they talking about the same same? Please clarify these aspects related to the spatial units taken into consideration in your analyses.
3) Can you write something more about the transaction data that you used? Have they been used in other analyses? I mean, Is there a literature concerning their use for analyses focused on regional and urban economic development? For example, it would be interesting to know something more about their advantages and (if so) disadvantages. I think that researchers from other countries would be interested in learning, critically, something more about the potential of these data.
Author Response
(1) Line 157: are three or two keywords? It seems that they are two: urbanization and economic recovery.
Reply:
The three key words I am referring to are COVID-19, urbanization patterns and economic recovery. I amended the sentence on line 157 to avoid misunderstanding the reader. Just as follows:
In view of this, this article focuses on three key words, namely the COVID-19, urbanization and economic recovery……
(2) How many cities, and how many regions are part of the Hubei province? Therefore, I wonder: how many cities are used as spatial units in the database? Commenting on table 2 (lines 232-239), authors talk about cities and regions as if they are the same spatial units. Are they talking about the same same? Please clarify these aspects related to the spatial units taken into consideration in your analyses.
Reply:
There are 17 cities in Hubei province, all of which are used as spatial units in the database. The first column in Table 2 is the ranking of cities. For example, in terms of total confirmed cases and total cured cases, Jingzhou ranks 4th among the 17 cities in Hubei Province, while Jingzhou ranks fifth and Ezhou ranks 4th in terms of total deaths.
(3) Can you write something more about the transaction data that you used? Have they been used in other analyses? I mean, Is there a literature concerning their use for analyses focused on regional and urban economic development? For example, it would be interesting to know something more about their advantages and (if so) disadvantages. I think that researchers from other countries would be interested in learning, critically, something more about the potential of these data.
Reply:
Thanks for the suggestion. In the third section of this article, we added a detailed description the data:
The unique data used in this paper is China UnionPay transaction data, which comes from the China UnionPay personal bank card database that includes cities issuing the cards, date and time of transactions, places where cards were physically swiped, names of stores, and amount of payment. In addition to traditional debit/credit cards swiping, the transaction method also includes mobile payment such as QR code scanning payment via QuickPass but does not include WeChat and Alipay QR code payment.
In addition to this article, we have also published other articles on the consumption network of the middle reaches of the Yangtze River and the overseas consumption of Mainland China, as well as some Chinese papers on the influence of consumption flows. The links of two English papers are as follows:
(1) Lei Wang, Wenyi Yang, Xiaoling Zhang, et al. Re-shaping global-ness by spending overseas: Analysis of emerging Chinese consumption abroad[J]. Cities, 2020,103034; https://doi.org/10.1016/j.cities.2020.103034.
(2) Lei Wang, Wenyi Yang, Yuan Yueyun, et al. Interurban Consumption Flows of Urban Agglomeration in the Middle Reaches of the Yangtze River: A Network Approach [J]. Sustainability, 2019,11(1), 268; https://doi.org/10.3390/su11010268.

Reviewer 3 Report
Authors conducted a detailed and exhaustive empirical analysis on the impact of current urbanization patterns on the spread of the epidemic and economic recovery from the perspective of time, industry and regional differences. This analysis is carried out in an orderly way and the reader can follow the rationale of the essay, also because of a very plain and easy English style and grammar. Therefore, the research results in a very interesting work for researchers, practitioners and for the general population.
However, I think it can be further improved by considering the following suggestions.
Introduction: at the end of the section, a paragraph on the rationale and contents of each section is missing. This is a pity, since a proper presentation would serve the cause of a better guide for the readers through the materials of the essay.
Line 58: a diagram of the curve might help to better visualize it.
Line 165: it would be useful to illustrate how the payment methods analyzed work. Are them mobile payments or do they include traditional debit/credit cards?
Author Response
(1) Introduction: at the end of the section, a paragraph on the rationale and contents of each section is missing. This is a pity, since a proper presentation would serve the cause of a better guide for the readers through the materials of the essay.
Reply:
Thanks for the suggestion. A description of the structure of the article has been added in the introduction. Just as follows:
The structure of the article is as follows: The second part of the article is literature review, which separately discusses the experience of urbanization and epidemics in Western society, as well as the background of COVID-19 in China and its impact on urbanization and economy. The third part is an introduction to data and methodology. The fourth part is the situation analysis of COVID-19, urbanization pattern and economic development in Hubei Province in 2020. Next, the fifth part analyzes the economic recovery effect of the urbanization pattern of Hubei Province under the impact of the COVID-19 from the perspective of time, industry and regional differences based on the results of the GMM models. Finally, the sixth part is the conclusion of the article, and puts forward suggestions for optimizing the urbanization pattern of Hubei Province and improving its ability to prevent epidemics.
(2) Line 58: a diagram of the curve might help to better visualize it.
Reply:
Thanks for the suggestion. We have revised this sentence as your request. Just as follows:
From a broader perspective, urbanization is an all-round change that includes economy, society, politics, culture, population and so on, which evolves along a diagram of the curve similar to the S-shape.
(3) Line 165: it would be useful to illustrate how the payment methods analyzed work. Are they mobile payments or do they include traditional debit/credit cards?
Reply:
Thanks for the suggestion. In the third section of this article, we added a detailed description the data:
The unique data used in this paper is China UnionPay transaction data, which comes from the China UnionPay personal bank card database that includes cities issuing the cards, date and time of transactions, places where cards were physically swiped, names of stores, and amount of payment. In addition to traditional debit/credit cards swiping, the transaction method also includes mobile payment such as QR code scanning payment via QuickPass but does not include WeChat and Alipay QR code payment.
